# Bot or Human? Detecting ChatGPT Imposters with A Single Question

**Hong Wang[1], Xuan Luo[1], Weizhi Wang[1], Melody Yu[2], Xifeng Yan[1]**
[1]University of California, Santa Barbara, CA, US
[2]Sage Hill School, Newport Coast, CA, US
{hongwang600,xuan_luo,weizhiwang}@ucsb.edu
ocmelodyyu@gmail.com, xyan@cs.ucsb.edu

## Abstract

Large language models (LLMs) like GPT-4 have recently demonstrated impressive capabilities in natural language understanding and generation. However, there is a concern that they can be misused for malicious purposes, such as fraud or denial-of-service attacks. Therefore, it is crucial to develop methods for detecting whether the party involved in a conversation is a bot or a human. In this paper, we propose a framework named **FLAIR**, **F**inding **L**arge Language Model **A**uthenticity via a Single **I**nquiry and **R**esponse, to detect conversational bots in an online manner. Specifically, we target a single question scenario that can effectively differentiate human users from bots. The questions are divided into two categories: those that are easy for humans but difficult for bots (e.g., counting, substitution, searching, and ASCII art reasoning), and those that are easy for bots but difficult for humans (e.g., memorization and computation). Our approach shows different strengths of these questions in their effectiveness, providing a new way for online service providers to protect themselves against nefarious activities. Our code and question set are available at https://github.com/hongwang600/FLAIR.

## 1 Introduction

Recently, the development of Large Language Models (LLMs) such as GPT-4 (OpenAI, 2023) and LLaMA-2 (Touvron et al., 2023a) has brought significant advances in natural language processing and achieved superior performance in downstream tasks of language understanding (Chowdhery et al., 2022), question answering (Su et al., 2019), dialogue systems (Wang et al., 2022b; Qian & Yan, 2023) and multimodal reasoning (Wang et al., 2022a). However, with the proliferation of these models, concerns have emerged regarding their potential misuse for malicious purposes. One of the most significant threats is the use of large language models to impersonate human users and engage in nefarious activities, such as fraud, spamming, or denial-of-service attacks. For instance, LLM agents could be used by hackers to occupy all customer service channels of various corporations, such as e-commerce, airlines, and banks. Moreover, with the help of text-to-speech (TTS) techniques, machine-generated voices could even occupy public service lines like 911, leading to severe public crises (Wang et al., 2021). These attacks could cause significant harm to online service providers and their users, eroding the trust and integrity of online interactions.

In response to these challenges, there is a pressing requirement for reliable differentiation between human users and malicious LLM-based bots. Conventional techniques, such as the use of CAPTCHAs (Von Ahn et al., 2003), have been developed to determine whether a user is a human or a bot in order to prevent bot spamming and raiding. A commonly used CAPTCHA method involves asking users to recognize distorted letters and digits. However, these approaches face significant challenges when it comes to detecting chatbots involving text only. This is where the emergence of large language models such as GPT-4 has further complicated the problem of chatbot detection, as they are capable of generating high-quality human-like text and mimicking human behavior to a considerable extent. Although recent

|  | Humans good at | Humans not good at |
|---|---|---|
| Bots good at | × | $\sqrt{}$ memorization computation |
| Bots not good at | $\sqrt{}$ symbolic manipulation randomness searching graphical understanding | × |

Table 1: Leveraging tasks that Bots and Humans are (not) good at.

studies such as DetectGPT (Mitchell et al., 2023) have proposed methods to classify if text is generated by ChatGPT or not, they focus on the offline setting. A recent study (Sadasivan et al., 2023b) shows that these detectors are not reliable under paraphrasing attacks, where a light paraphraser is applied on top of the generative text model. This limitation highlights the need for more robust and accurate methods to differentiate large language models from human users and detect their presence in online chat interactions.

In this paper, we propose a novel framework named **FLAIR**, Finding LLM Authenticity with a Single Inquiry and Response, to take full advantage of the strength and weakness of LLMs for LLM-based conversational bot detection. Specifically, we introduce a set of carefully designed questions that induce distinct responses between bots and humans. These questions are tailored to exploit the differences in the way that bots and humans process and generate language. As shown in Table 1, certain questions in the fields of symbolic manipulation, randomness, searching, and graphical understanding are difficult for bots but relatively easy for humans. Examples include the counting, substitution, searching, and ASCII art reasoning. On the other hand, memorization and computation was relatively easy for bots but difficult for humans.

Our experimental results demonstrate that FLAIR provides a viable alternative to traditional CAPTCHAs. Specifically, while humans and LLMs excel with high accuracy on tasks within their areas of strength, their performance significantly declines on tasks they are less adept at, often falling to very low levels ($\sim$0%). This sharp disparity in performance allows for the differentiation between human and LLM respondents with just a single question. The proposed approach shows promise in developing more robust and accurate methods to quickly detect bots and safeguard online interactions.

## 2 Related Work

### 2.1 CAPTCHA

CAPTCHA (Von Ahn et al., 2003) is a common technique used to block malicious applications like dictionary attacks, E-mail spamming, web crawlers, phishing attacks, etc. There are different types of CAPTCHAs. Text-Based CAPTCHAs require the users to recognize letters and digits in distortion form (Chew & Baird, 2003; Mori & Malik, 2003; Yan & El Ahmad, 2008), while Image-Based CAPTCHAs (Gossweiler et al., 2009) require users to choose images that have similar properties such as traffic lights. Video-Based CAPTCHAs (Kluever, 2008) require the user to choose three words that describe a video, and Audio-Based CAPTCHAs (Gao et al., 2010) ask the user to listen to an audio and submit the mentioned word (Saini & Bala, 2013). Puzzle CAPTCHAs (Singh & Pal, 2014) require the user to combine segments to form a complete picture. These techniques are used to differentiate between human users and bots, preventing malicious activities.

### 2.2 LLM offline detection.

Since its introduction, Large Language Models (LLMs) have become widely used and raised public concerns about potential misuse. For instance, students may use ChatGPT (OpenAI, 2023) to complete written assignments, making it difficult for instructors to accurately

assess student learning. As a result, there is a growing need to detect whether a piece of text was written by ChatGPT. To tackle this problem, DetectGPT (Mitchell et al., 2023) proposes a solution by comparing the log probabilities of the original passage with that of the perturbations of the same passage. The hypothesis behind this method is that minor rewrites of text generated by the model would likely result in lower log probabilities compared to the original sample, while minor rewrites of text written by humans may result in either higher or lower log probabilities. Another line of study model this problem as binary classification problem and fine-tune another model using supervised data (Bakhtin et al., 2019). Most recently, Mitrović et al. (2023) fine-tunes a Transformer-based model and uses it to make predictions, which are then explained using SHAP (Lundberg & Lee, 2017). Another area of research focuses on adding watermarks to AI-generated text in order to facilitate their identification, which involves imprinting specific patterns on the text to make it easier to detect (Zhao et al., 2023). Soft watermarking, as proposed by Kirchenbauer et al. (2023), involves dividing tokens into green and red lists in order to create these patterns. When generating text, a watermarked LLM is more likely to select a token from the green list, which is determined by the prefix token. These watermarks are often subtle and difficult for humans to notice.

However, as demonstrated in Sadasivan et al. (2023a), a range of detection methods, including watermarking schemes, neural network-based detectors, and zero-shot classifiers, can be easily defeated by paraphrasing attacks. These attacks involve applying a light paraphraser to text generated by a language model. Furthermore, a theoretical analysis suggests that even the best possible detector can only perform marginally better than a random classifier when dealing with a sufficiently good language model. This highlights the fundamental challenge in offline detection of text generated by advanced language models, which can produce writing that is virtually indistinguishable from human-written text. Thus, it is more meaningful and crucial to shift the focus to online detection settings where users engage in live chat interactions with the system.

## 3   Leveraging the Weakness of LLM

In this section, we explore specific tasks such as Counting, Substitution, Random Editing, Searching, and ASCII Art Reasoning. These tasks, while seemingly straightforward for humans, present significant challenges for large language models (LLMs).

### 3.1   Counting

State-of-the-art LLMs cannot accurately count characters in a string (Qian et al., 2022), while humans can do so with ease. This limitation of LLMs has inspired the design of a counting FLAIR to differentiate humans and LLMs. Participants are asked to count the number of times a specific character appears in a given string:

> Answer the question without explanation: Please count the number of t in eeooeotetto
> GPT-3.5: There are 4 t's in "eeooeotetto".
> Human: 3

As demonstrated by this example, GPT-3.5 struggles to provide an accurate count of the specified character within the string.

### 3.2   Substitution

It is known that LLMs often output contents that are inconsistent with context (Elazar et al., 2021; Wang et al., 2022c). It is a shared weakness of current LLMs. We ask LLMs to spell a random word under a given substitution rule, testing if they can follow the rule consistently. The random word has a length between five to ten characters, which is randomly sampled from a dictionary. Here is an example:

Answer the question without explanation: Use m to substitute p, a to substitute e, n to substitute a, g to substitute c, o to substitute h, how to spell peach under this rule?
GPT-3.5: mnong
Human: mango

LLMs are struggle with applying this rule consistently and repeatedly as in the above substitution task. This testing concept could be further generalized to encryption scheme where a string is transformed based rules.

### 3.3 Random Editing

Random editing is a technique used to evaluate the robustness of natural language processing models against noisy inputs. The technique involves presenting the model with a string of characters (sampled from the set of $[0, 1]$) and randomly dropping, inserting, swapping, or substituting some characters. To verify that the model is capable of performing these random operations, it is required to generate three different outputs under the same conditions. By ensuring that the three outputs are different, we can confirm that the model is applying the random operation.

For random drop, the model is asked to randomly remove $k$ occurrences of a given character $c$ from the string, where $k$ is a random number and $c$ is a random character that appears in the string. For random insert, the model is asked to randomly insert $k$ occurrences of a given character $c$ into the string at random positions. For random swap, the model is asked to randomly swap $k$ pairs of characters $c$ and $d$ in the string, where $c$ and $d$ are different characters that appear in the string. Finally, for random substitution, the model is asked to randomly replace $k$ occurrences of character $c$ with another character $d$ in the string. Below we show an example of LLMs and human output on the random drop task:

Answer the question without explanation: randomly drop two 1 from the string: 0110010011. Give me three different outputs.
GPT-3.5: 01001011, 010010011, 011001011
Human: 00010011, 01000011, 01100001

While humans can intuitively and correctly manipulate characters as specified, GPT-3.5 struggled to consistently perform the task correctly. This suggests a potential limitation in handling dynamically specified, random input transformations in natural language processing tasks.

### 3.4 Searching

Searching is an inherent ability of humans, allowing us to envision various potential outcomes in our minds and make decisions accordingly. In contrast, language models are constrained by their sequential generation process, which limits their ability to explore different results through backtracking. Once a token is generated, these models cannot retract or revise it based on subsequent insights or information, preventing them from exploring alternative paths that might have been overlooked during initial processing. We test this by examining a fundamental search scenario: counting the number of islands in a 2D map, a task typically solved using Depth-First Search (DFS) or Breadth-First Search (BFS) algorithms. This challenge requires identifying and counting the connected components of '■'s (representing land) in a grid. Adjacent '■'s, either horizontally or vertically, form an island. To accurately determine the number of islands, one must systematically traverse the map, marking visited land cells to avoid recounting the same island. Here is an example:

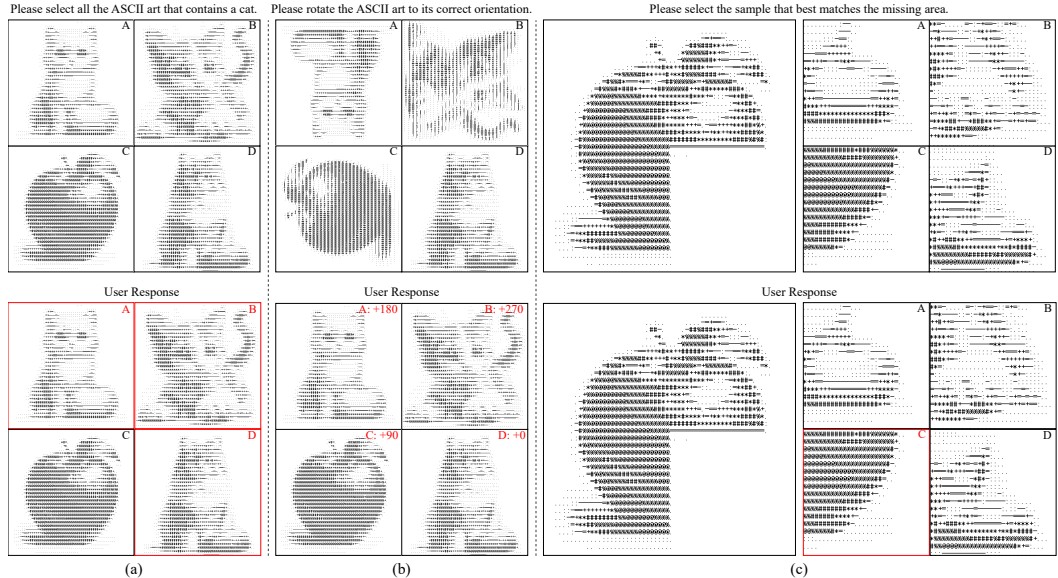

Figure 1: Example about ASCII reasoning. (a) select the ascii arts containing X. (b) Rotate the ASCII art to the appropriate orientation. (c) Select the one that most accurately aligns with the cropped portion.

Answer the question without explanation: Count the number of islands in a given 2D map, where black blocks represent land and spaces represent water. Output a single number as your result.
Map:

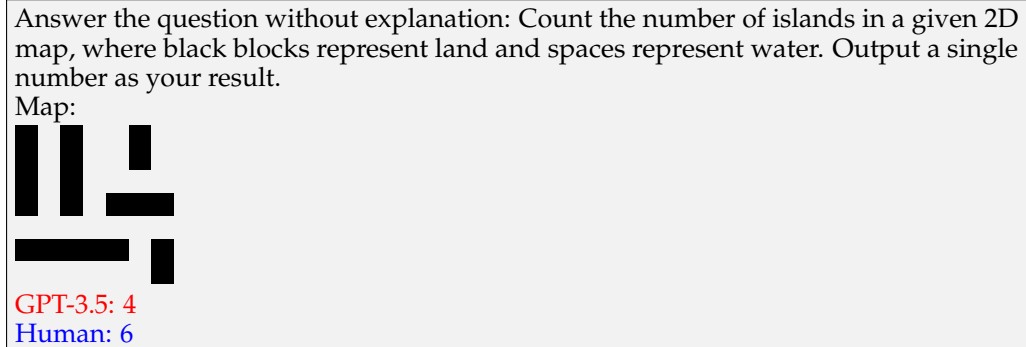

GPT-3.5: 4
Human: 6

As demonstrated in the example, GPT-3.5's output is incorrect, while humans can intuitively and accurately count the number of islands by traversing the connected areas. This task highlights the inherent limitations of language models in search and backtrack mechanisms.

## 3.5 ASCII Art Reasoning

Humans can comprehend visual elements whether they are presented in image format or as text strings. For example, an apple converted into ASCII art can still be recognized by people, despite the abstract representation. This ability to understand ASCII art requires a level of visual abstraction that current language models lack. To investigate this, we converted natural images into ASCII format and conducted experiments involving common reasoning tasks.

To integrate this approach with existing CAPTCHA systems, we focused on three reasoning tasks. The first task involves identifying ASCII art that includes a specific element, such as a cat. Figure 1(a) presents an example where participants are required to select the ASCII art depicting a cat, mimicking the function of a reCAPTCHA. Participants click on images containing a cat. For language models, they receive these images in string format with designated indices (A, B, C, D). The language model must then identify and output the indices corresponding to ASCII art containing a cat. Below is the prompt used for Figure 1(a), where the [ASCII art X] represents the strings of the ASCII arts:

> Without providing explanations, select all ASCII art that contains a cat. Output the indices of the ASCII arts.
> A: [ASCII art A] B: [ASCII art B] C: [ASCII art C] D: [ASCII art D]
> GPT-4: The ASCII arts containing a cat are in options A and C.
> Human: A, B, D

The answers for this example are $[A, B, D]$, which are colored in red in Fig 1 (a). The second task involves rotating the ASCII art to the correct direction. Fig 1 (b) shows an example. Similar to the image-based rotation CAPTCHA, participants are required to use a mouse to drag and rotate the image to the correct direction. For this task, we will also prompt the LLM with ASCII art but require the LLM to output the degree needed to rotate clockwise for each image. The answers for this example are: $[A : +180, B : +270, C : +90, D : +0]$. The third task is to choose one ASCII art that best fits the cropped part. Fig 1 (c) demonstrates an example. The LLM has to output the index of the ASCII art that best fits. We can also define other more complicated visual reasoning tasks like "Please click the images containing a cat, plane, keyboard, banana in order," or "Please combine the 4 parts into a single image," etc. This method is compatible for many existing CAPTCHAs that does not require detailed reasoning (convert image to ASCII arts will lose the details), and can be seamlessly combine with many existing systems.

Our tests indicate that ASCII-based reasoning tasks present substantial challenges for current Large Language Models (LLMs), including GPT-4. Despite utilizing techniques like chain-of-thought reasoning or deploying Python APIs, these models face difficulties. We further extended these tests to more advanced visual language models, such as GPT-4o, and found that they are limited to handling only the simplest of ASCII reasoning task.

## 4 Leveraging the Strength of LLM

In this section, we will discuss the methods that capitalize on the strengths of LLMs. These questions are typically challenging for humans, but are relatively easy for LLMs due to their ability to memorize vast amounts of information or perform complex computation.

### 4.1 Memorization

The primary idea behind this type of questions is to ask users to enumerate items within a given category. Below is an example question and answers from GPT-3.5 (we only list a few items from the output):

> Answer the question without explanation: List the capitals of all the states in US:
> GPT-3.5: 1. Montgomery - Alabama 2. Juneau - Alaska 3. Phoenix - Arizona ... 50. Cheyenne - Wyoming
> Human: I don't know.

For humans, this type of question is not easy as it requires a good memorization. There are several choices when designing enumerating questions. The first option is to contain many items to make it more challenging for humans to remember, such as all the countries in the world. The second option is to include relatively old information that people may not have encountered, such as all the movies in 1970s. The third option is domain-specific information that people are unlikely to know, such as the names of all Intel CPU series. We can determine the likelihood of the answer coming from a bot by verifying the overlap between the given answer and the ground truth. If the overlap is greater than a threshold, then it is more likely that the answer comes from a bot.

### 4.2 Computation.

Performing complex calculations, such as multiplication, without the aid of a computer or external notes is difficult for humans due to the challenges of recalling intermediate steps. In contrast, LLMs excel in remembering the results of common equations, such as the square of $\pi$. For instance, below is an example question and the answer generated by GPT-3.5:

> What is the square of $\pi$?
> GPT-3.5: The square of $\pi$ (pi) is approximately 9.869604401.
> Human: I don't know.

Moreover, by utilizing external tools, such as Wolfram, GPT Plugins can even solve more intricate problems with in a second. But for human, it will require a much longer time to solve this problem. Therefore, the behavior of real human will be very different from a language model and we can easily detect bots using a behavior-based CAPTCHA system (Von Ahn et al., 2008; Awla et al., 2022; Guerar et al., 2021; Acien et al., 2021).

# 5    Experiments

In this section, we present the experimental results of our proposed single questions for distinguishing between humans and LLMs. We curate a dataset for each category of the proposed questions, which is used to evaluate the performance of both humans and LLMs. By contrasting the accuracy of responses, we aim to differentiate between the two.

## 5.1    Datasets

To evaluate the performance of both LLMs and humans, we constructed a dataset for each category of questions and open-sourced it on `https://github.com/hongwang600/FLAIR`.

**Counting**    We used the entire alphabet for this task. First, we randomly picked one letter to be the target letter to count. Then, we chose a random number $k$ between 10 and 20, which determined how many times the target letter would appear in the string. We created a string that was 30 characters long, with $k$ instances of the target letter and the remaining $30 - k$ characters randomly selected from the other letters of the alphabet. The correct answer for the number of times the target letter appeared in the string was $k$, and an answer matching this number was considered correct.

**Substitution**    To create our dataset, we began by collecting the top 1500 nouns from the Talk English web site[1]. We then filtered the words to include only those with a length between 5 and 10 characters. Next, we randomly generated 100 pairs of words, each with a corresponding substitution map that could transform one word into the other. To ensure the validity of our pairs, we excluded any that would require one character to be mapped to more than one character, which would result in a conflict. The resulting questions presented to participants included the substitution rule and the original word, with the answer requiring another word produced through the substitution.

**Random Editing**    In the random editing task, we evaluate the model's ability to perform four distinct operations: drop, insert, swap, and substitute, as described in Section 3.3. For each operation, we generate a random binary string of length 20 to ensure readability. We randomly sample parameters such as the target character and the operation count. Participants are then asked to produce three different outputs after performing the specified random operation. For the random drop task, participants are required to randomly remove $k$ occurrences of a given character $c$ from the string. In the random insert task, participants randomly insert $k$ occurrences of a given character $c$ into random positions within the string. For the random swap task, participants randomly swap $k$ pairs of characters $c$ and $d$ within the string. Lastly, in the random substitution task, participants randomly replace $k$ occurrences of character $c$ with another character $d$. To validate the correctness of the outputs, we first check each individual output by comparing it to the original string. We then verify that the three outputs differ from each other. An answer is considered correct only if each output is correct and all three outputs are distinct. This comprehensive validation ensures the model's ability to handle dynamically specified, random input transformations accurately.

---

[1]website URL: `https://www.talkenglish.com/vocabulary/top-1500-nouns.aspx`

**Searching**    We generated 100 random 7x7 grids containing spaces and ■s to create a map. Each cell in the grid was independently filled, with a 50% chance of placing a 1 and a 50% chance of placing a 0. While filling the grid, we ensured that no cell would be filled on the diagonal of an already filled land cell, thereby avoiding diagonal connections. The filling process was conducted one cell at a time, with each new cell sampled from the remaining available spaces. We then utilized the Depth-First Search (DFS) algorithm to determine the number of islands, which served as the ground truth. An answer is considered correct if it matches this ground truth.

**ASCII Art Reasoning**    In our study, we employed GPT-4 to create a list of four items, incorporating two random entities, such as [A: cat, B: cat, C: apple, D: cat], as illustrated in Figure 1. These entities served as prompts for Dall-E to produce correspondent images. Subsequently, we utilized the methodology outlined in "ASCII Art Reasoning" to transform these images into ASCII art. Each piece of ASCII art was rendered in a 64x64 character matrix using an ASCII gradient of '@%#*+=-:. '. For the task "select the ASCII arts containing X", we chose one entity (e.g., cat) at random and use the generated ASCII arts to form the prompt for language model. In the task "Rotate the ASCII art to the appropriate orientation", we randomly rotated the ASCII art pieces by 90, 180, 270, or 360 degrees clockwise and used these variations as prompts for the language model. For the task "select the one that most accurately aligns with the cropped portion", we randomly selected one piece of ASCII art and cropped a random 1/4 portion from its top-left, top-right, bottom-left, or bottom-right corner. We then cropped three additional patches from different images at the same positions. Fig 1 (c) is an example. The language model was prompted to identify the patch that best matched the original cropped portion. For these three tasks, the answer indices/degrees from language model should be exactly the same with the ground truth to be considered correct.

**Memorization**    We used a set of questions that required the user has a good memorization. There are two types of question under this category including enumerating and domain-specific questions. For enumerating, the user is asked list items within a given category. We manually collected 100 categories containing more than 50 items or those that were difficult for humans to know with the help of ChatGPT. The question asked users to list the items within the given category, and we calculated the coverage of the response against the ground truth. If the coverage exceeded the threshold of 95%, we considered the answer to have been generated by an LLM. For domain specific questions, we manually collected a set of 100 questions whose answers are difficult for people to recall or access, such as "What is the weight of the world's heaviest lemon in grams?". Although these questions may be challenging for humans to answer, they are relatively easy for large language models (LLMs) due to their pre-training on large corpora that includes these questions. All of the reasonable results that can be sources on the internet are considered correct.

**Computation**    To create the computation dataset, we selected the problem of four-digit multiplication. Specifically, we randomly sampled 100 pairs of four-digit numbers and calculated their product as the ground truth. Participants were asked to solve these multiplication problems and were considered correct if absolute difference between their answer and the ground truth was within 10%. For humans, it can be difficult to accurately calculate these multiplications without the aid of notes or a calculator, leading them to often respond with "I don't know". In contrast, large language models (LLMs) have seen many similar equations during pre-training and tend to provide a guess that is often close to the ground truth. This testing can be further extended to any complicated computation like division, exponents, etc.

## 5.2  Benchmarking Baselines

We conducted experiments across a range of LLMs, including open-source models like Vicuna-13b (Chiang et al., 2023) and LLaMA-2-13b, 70b (Touvron et al., 2023b), as well as proprietary models such as GPT-3, 3.5, and 4 (Brown et al., 2020; OpenAI, 2023; Ouyang et al., 2022). Additionally, we experimented with the zero-shot chain-of-thought (CoT)

methodology (Wei et al., 2022), where GPT-4-CoT denotes the results obtained using this specific approach with GPT-4. To implement this, we appended a structured prompt to the standard query, instructing the model to "Please analyze the question step by step and output your analysis under 'Analysis', then provide your final answer based on this analysis after 'Answer:'". We then extracted the part after 'Answer:' to verify the result's correctness.

For tasks requiring computational solutions, we used a variant called GPT-4-py. Here, GPT-4 was instructed to compose Python code with the prompt "Please write a python program to solve this problem". The generated code was executed through an API call to derive the final outcome. We compared the code's output with the ground truth to verify correctness. Code that could not be successfully executed or did not produce any output was considered incorrect.

## 5.3 Main Results

We conducted a series of five trials for the LLM tests, with each trial consisting of 100 samples, matching the sample size used in the user study. To determine the final accuracy, we used the median accuracy from these five trials. Tasks that language models struggle with tend to exhibit higher variance in accuracy due to the instability of their outputs. This instability introduces randomness, leading to potential outliers. By using the median accuracy, we minimize the influence of these outliers and provide a more robust measure of performance.

For our user study, we enlisted 10 participants. The age distribution among our participants was varied, comprising one individual aged between 10-20 years, five aged between 20-30 years, three aged between 30-40 years, and one aged between 40-50 years. Each participant responded to a set of 10 questions in each category, with a 10-second time limit per question. This time limit prevents users from searching the internet or using a calculator to solve the computation and memorization tasks. For the random editing and ASCII-related tasks, we extended the time limit to 30 seconds. The results of user study is shown in the row "Humans".

| | Count | Substi. | Random Edit | Search. | ASCII Select | ASCII Rotate | ASCII Crop | Memori. | Comput. |
|---|---|---|---|---|---|---|---|---|---|
| Humans | 100% | 100% | 100% | 100% | 100% | 97% | 94% | 6% | 2% |
| Vicuna-13b | 15% | 1% | 0% | 1% | 3% | 0% | 17% | 93% | 100% |
| LLaMA-2-13b | 10% | 2% | 0% | 2% | 4% | 1% | 23% | 94% | 96% |
| LLaMA-2-70b | 14% | 5% | 3% | 2% | 4% | 0% | 22% | 97% | 98% |
| GPT-3 | 13% | 2% | 0% | 0% | 2% | 0% | 19% | 94% | 95% |
| GPT-3.5 | 15% | 6% | 2% | 3% | 5% | 1% | 26% | 99% | 98% |
| GPT-4 | 21% | 8% | 6% | 7% | 6% | 0% | 25% | 99% | 99% |
| GPT-4-CoT | 33% | 56% | 13% | 10% | 2% | 0% | 27% | 99% | 99% |
| GPT-4-py | 100% | 100% | 100% | 100% | 0% | 0% | 0% | N/A | N/A |

Table 2: The comparison between LLMs and Human on different tasks.

The results, presented in Table 2, align with the tasks outlined in Section 3, "Leveraging the Weaknesses of LLMs" (left side), and Section 4, "Leveraging the Strengths of LLMs" (right side).

In the left side, we evaluate human versus LLM performance on tasks that are straightforward for humans but challenging for LLMs. Humans attained perfect scores (100%) across most tasks, with slight variations in the ASCII Art tasks, where the resolution of some ASCII representations posed recognition challenges. Conversely, LLMs struggled significantly with tasks involving substitution, random edits, searching, and ASCII art, often achieving near 0% accuracy. LLMs showed improved performance on tasks like counting and ASCII Crop, benefiting from a more constrained solution space which simplified the task of identifying correct answers. For example, with only four possible answers (A, B, C, D) in the ASCII Crop task, the chance of selecting the correct one is approximately 25%.

The results also demonstrate that larger and newer language models generally have better performance on these tasks. GPT-4, which has 1.8T parameters, exhibits superior performance compared to Vicuna-13b on nearly all the tasks. We also explored advanced techniques like chain-of-thought prompting (Wei et al., 2022). The GPT-4-CoT results show the performance of GPT-4 with chain-of-thought prompting. The experiments demonstrated that chain-of-thought prompting can improve performance on tasks that can be decomposed into several steps, especially for substitution, where the language models can perform substitution step-by-step. However, chain-of-thought prompting does not help to improve performance on searching and ASCII-based visual reasoning tasks. Its effectiveness is limited to textual and decomposable tasks like counting, substitution, and random edit.

Furthermore, we explored the possibility of GPT-4 generating code to solve the problems by manually prompting it with, "Please write a Python code to solve the problem." With this human-assisted prompt, GPT-4 can generate code that can bypass the challenges in counting, substitution, random edit, and searching tasks, achieving 100% accuracy. However, GPT-4 is not intelligent enough to write code for ASCII art reasoning. The attempts often result in erroneous or non-functional code that does not output anything, yielding a 0% accuracy rate across ASCII reasoning tasks. This suggests that with manual prompting on the use of appropriate tools, language models can overcome some of their limitations.

The right section of Table 1 focuses on tasks that are challenging and time-consuming for humans but relatively straightforward for LLMs. The results reveal that humans struggled with tasks that require exceptional memory or computational abilities, with only 6% and 2% of human participants successfully completing the memorization and computation tasks, respectively. In stark contrast, LLMs demonstrated remarkable proficiency in these areas, with some models achieving near-perfect accuracy.

### 5.4 Discussion

The rapid advancements in LLMs pose a significant challenge to the long-term effectiveness of our proposed approach. It is conceivable that future multimodal language models could automatically identify tasks requiring code generation or visual reasoning, potentially undermining our method. Therefore, it is crucial to continuously monitor the development of LLMs and adapt our approach accordingly.

One promising direction is to integrate our method with existing frameworks, such as behavioral CAPTCHAs, which take into account users' clicks, keyboard operations, time usage, and other behaviors. Combining these techniques can provide a more robust defense against increasingly sophisticated bots.

During our research, we also explored a new set of questions that require few-shot reasoning, which current language models struggle to solve but might be too challenging for the general public. For example, consider the question: "I have a set of strings following a certain pattern ["abcde|||||edcba", "ab||ba", "abc|||cba", ...]. What is the string that contains exactly 4 |?" Humans can intuitively find the pattern by identifying the consecutive characters and their reversals. However, without additional training, language models often fail to provide the correct answer, even when prompted to write Python code. Although language models are claimed to be few-shot learners, they have limited generalization to new tasks without specific training. The direction of discovering few-shot questions remains promising and could lead to new methods for bot detection.

## 6 Conclusion

In conclusion, this paper proposes a new framework called FLAIR for detecting conversational bots in an online environment. The proposed approach targets a single question scenario that can effectively differentiate human users from bots by using questions that are easy for humans but difficult for bots, and vice versa. Our experiments demonstrate the effectiveness of this approach and show the strengths of different types of questions. This framework provides online service providers with a new way to protect themselves against fraudulent activities and ensure that they are serving real users.

## Acknowledgment

We would like to thank the anonymous reviewers for the helpful comments. We extend our gratitude to Ethan Mader for his insightful discussions. This research was partly supported by the DARPA PTG program (HR001122C0009) and a generous gift grant from Visa Research. The opinions, findings, conclusions, and recommendations presented in this paper are those of the authors and do not necessarily represent the views of the funding agencies.

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
