# OpenReview forum: "Bot or Human? Detecting ChatGPT Imposters with A Single Question"
_colmweb.org/COLM/2024/Conference — COLM_

### Official Review · Reviewer_SB6E · 2024-05-10

**Rating:** 6
**Confidence:** 4
**Ethics Flag:** 1

**Summary:**

The paper evaluates a number of tasks on several LLMs and on humans that are a priori selected and experimentally confirmed (in most instances) to be either very hard for LLMs and easy for humans or the other way round. The main motivation put forward by the authors is to identify ways to perform one-shot classification between human and artificial conversational agents.

**Questions To Authors:**

- The most interesting tasks being ASCII-art-related, how do you relate your results on these tasks with the ArtPrompt paper by Jiang et al. 2024 (published after the submission deadline), where the authors seem to show that LLMs do understand ASCII-art to some extent?

**Reasons To Accept:**

- The paper is based on a simple yet relevant idea that the authors investigate in a simple yet effective way

**Reasons To Reject:**

- The impact of the work is not sufficiently discussed. The authors write that "overcoming [the poor level of GPT-4 ASCII-art related capabilities] would necessitate significant investment, potentially amounting to millions of dollars, for developing a dedicated ASCII art reasoning model". I am not convinced by this statement at all. GPT-4 is very good at multiple tasks for which no "dedicated reasoning model" was developed. And the "millions of dollars" ballpark is not motivated at all. On the other hand, the authors do not mention the fact that a malevolent LLM practitioner might try to make their LLM bad at memorization and computation tasks, in order to behave in a more human-like fashion.
- The paper needs to be proofread. It is regrettable for a paper to be submitted with typos like the following, all easily detected by any good spell checker (including Microsoft Word or… ChatGPT):
    - page 3 "another line of study model this problem" ("models")
    - page 4 "an example of LLMs and humans output" ("an example of LLM and human output")
    - page 4 "As we can seen" ("see")
    - page 4 "This first stage of our proposed method is convert" ("is to convert")
    - page 5 "The next step is perform" ("is to perform")
    - page 6 "Below is and example" ("an")
    - page 6 "it require a good memorization" ("requires")
    - page 8 "10 seconds time limit" ("10-second time limit", with hyphen but no -s)
    - page 9 "develope" ("develop")
    - page 9 "possivle" ("possible")
    - page 9 "to Combine" ("to combine")
    - some citations should have been made using \citep and not \citet (examples: all citations in section 5.2, "Awla et al. (2022); Guerar et al. (2021)" on page 9)

---

> ### Author Rebuttal · Authors · 2024-05-31
>
> We appreciate your thorough review of our paper. Your feedback has prompted us to revisit and clarify several aspects of our work.
>
> **Impact of the work**: Regarding the impact of our work, we acknowledge your concerns about the statement regarding the potential investment required for developing a dedicated ASCII art reasoning model. We agree that the estimate provided may not have been adequately justified. While it is true that GPT-4 demonstrates proficiency in multiple tasks without dedicated models, our intention was to highlight the specific challenges posed by ASCII art reasoning and other questions we identified (the newest one is “Can we make 24 using only four 7's?”), which may require significant research and development efforts to overcome. We will revise the manuscript to provide a more nuanced discussion on this point, taking into account the broader context of AI model capabilities.
>
> Your observation about the potential for malevolent LLM practitioners to manipulate model behavior to mimic human-like performance is insightful. We understand the concern, and it is indeed possible for dedicated solutions to address specific tasks. However, the proposed FLAIR method involves a set of detection questions and will evolve with the findings of more possible detection questions. It is still quite hard for advanced LLMs or Multimodal LLMs to overcome the complicated tasks like ASCII-Art recognitions. The LLM imposters may overcome some detectors like memorization questions but they are still easy to be detected via a combination of multiple detection questions.
>
> **ArtPrompt Paper**: We appreciate your reference to the ArtPrompt paper by Jiang et al. (2024). That paper focuses on leveraging ASCII art to bypass safety checks and invoke undesired responses from LLMs. It demonstrates that LLMs can recognize simple ASCII art representations of words, i.e., the characters of "BOMB". In contrast, our work shows that LLMs still struggle with more complex ASCII art tasks, especially when manipulations like cropping and rotations are involved. Nevertheless, we will cite this related work and discuss it.
>
> **Typos**: We sincerely apologize for the typos we made in the manuscript. We will rectify these errors accordingly.

---

> > ### Comment · Reviewer_SB6E · 2024-06-05
> >
> > Thank you for your reply. I appreciate the fact that you acknowledge my concerns about poorly motivated statements. I also appreciate your reply to my comment on possible malevolent LLM practitioners, but I remain unconvinced. My comments on the matter are actually indirectly related to the second "reason to reject" put forward by reviewer fBF2, which are also very relevant. Unfortunately, I do not feel that your answers to these comments are convincing enough to justify a change in my overall rating.

---

### Official Review · Reviewer_mR8r · 2024-05-11

**Rating:** 6
**Confidence:** 3
**Ethics Flag:** 1

**Summary:**

This paper introduces a novel method to detect whether a participant in a conversation is a human or a bot. Specifically, they evaluated various question-answer pairs on both humans and LLMs. These were single QA pairs that included questions about counting, substitution, random editing, ASCII art reasoning, memorization, and computation. The questions are designed around things that the LLMs are good at (e.g., memorization and computation) and bad at (i.e., other tasks). Overall, the paper finds that ascii-related questions can accurately be used to detect if someone is a bot or not. Specifically,  only 1% of questions were accurately answered on the ASCII rotate questions for some models, but 97% were accurate for humans. Hence, these systems can potentially be used by companies and online services to mitigate bot accounts.

**Questions To Authors:**

- Why median accuracy vs. mean accuracy?
- How much did the accuracy vary from trial to trial?

**Reasons To Accept:**

- Overall, this is a really unique study that explores ways to detect bots in the wild (e.g., online service providers) so they can't cause issues.
I-  definitely want to highlight the creativity of the ASCII solution. This is really out-of-the-box thinking.
- The paper is generally well-written.

**Reasons To Reject:**

- The paper would be improved if there was a broader discussion about frameworks for how new questions can be generated as LLMs overcome these limitations. If the questions require manual intervention, they will not necessarily scale at the speed needed to mitigate attacks. While the findings are exceptionally creative, it does seem a bit early in the analysis to understand the impact of the findings. Could there be a general automated way to generate these questions that target weaknesses? If so, how would it be done? At a minimum, if substantially more work is required to do automated question-answering, this should be mentioned as a limitation.
- More description about how accuracy was calculated. There are also some general questions that I have, e.g., why median accuracy vs. mean accuracy? How much did the accuracy vary trial-to-trial?
- [Minor Weakness] I'm not entirely sure about the usefulness of this system in the application for VLLMs. Particularly, what would stop a bot from screenshotting a website, performing a rotation on the image, then converting back to ascii art? This is possible via GPT4 online interface based on superficial things I tested (e.g., it generates code that converts an image to ASCII and creates code that can perform rotations on the image). I assume a similar thing can be done for matching to images, e.g., converting the ASCII art to an image and then doing image-to-image matching.

---

> ### Author Rebuttal · Authors · 2024-05-31
>
> Thank you for your thoughtful and detailed review of our paper. We appreciate the suggestions you raised in your review and will revise the paper accordingly.
>
> **Broader Discussion on Frameworks for New Question Generation**: Our current practice is to identify different classes of questions that LLMs can not handle very well (or easy for LLMs, but hard for human).  For most of these classes, questions/answers are indeed generated automatically.  For the memorization task,  we manually picked up 100 questions. However, this can be done automatically by feeding questions to GPT and checking the cardinality of their answers (possibly by GPT again).  We will give a thorough discussion on scaling the generation of question/answer pairs in the revision.
>
> The identification of the question class requires substantial research and experimentation. This remains a fertile area for future research, aimed at identifying the true weakness of LLMs. We will incorporate this discussion into our revised manuscript to highlight the importance of ongoing research in this area.
>
> **Details on Accuracy Calculation**: In the original submission, we reported the median accuracy values across 5 trials for each task and model, with each trial involving 100 samples. We acknowledge that providing the mean accuracy values could offer additional insights. Below, we present a portion of the results, showing the mean accuracy values for LLaMA-2-70b, GPT-3.5, and GPT-4. The values within the brackets represent the variance across the 5 trials:
>
> |Model|Count|Substi.|Random Edit|ASCII Select|ASCII Rotate|ASCII Crop|Memori.|Comput.|
> |--------|-------|---------|-----|------|-------|--------|-------|-----|
> |LLaMA-2-70b|13.8% (4.16)|5.4% (2.63)|2.2% (2.16)|4.2% (2.56)|0.6% (0.64)|23.0% (5.6)|96.8% (2.96)|97.8% (2.55)|
> |GPT-3.5|15.2% (2.56)|5.4% (4.24)|1.8% (1.36)|5.0% (2.00)|1.4% (0.24)|25.4% (4.64)|98.4% (1.44)|98.0% (1.20)|
> |GPT-4|21.0% (6.00)|8.2% (4.16)|6.4% (2.63)|5.6% (3.44)|0.6% (0.64)|24.8% (4.16)|99.2% (0.56)|99.0% (0.40)|
>
> **Concerns about Practical Application and VLLMs**: We apply the latest VLLM, GPT-4O on ASCII-art tasks. However, GPT-4o's performance on ASCII-art tasks is still limited, reaching only 0% accuracy for rotation subtask and 37% accuracy for cropping. These results suggest that while latest visual language models do possess some visual reasoning capabilities, the proposed advanced tasks like ASCII-art are still effective in imposter detections.

---

> > ### Comment · Reviewer_mR8r · 2024-06-04
> > **Comment**
> >
> > Thanks for your response.
> >
> > This clarifies many of my questions. Regarding "The identification of the question class requires substantial research and experimentation," this should be discussed more as a limitation, particularly if methods can easily be trained specifically to overcome these tasks that were presented.

---

### Official Review · Reviewer_fBF2 · 2024-05-14

**Rating:** 5
**Confidence:** 4
**Ethics Flag:** 1

**Summary:**

This paper propose a new CAPTCHAs methods to detect whether the response is from human or  LLMs via asking a question that is either easy to LLM but hard to human or easy to human but hard to LLM. The researcher identify a few patterns of such questions that can lead to distinguished performance of human and LLMs. The proposed questions are checked against human users and a comprehensive set of LLMs. The results show that these questions can effectively distinguish the LLMs and Humans based on the accuracy of their response;

**Reasons To Accept:**

1. A simple and clean design to capture the different behaviors from human users and LLM via asking a question that will lead to completely different response from the two groups.
2. Details of the question design are clearly explained which makes it easy to replicate the dataset by following reseaches.
3. The benchmark is also valuable to check LLM's capability on the founda-mental skills that is easy for human

**Reasons To Reject:**

1. It is questionable on the time-wise effectiveness of this approach to distinguish LLM from human as we don't know whether the challenges faced by LLMs now will still remain when newer version of LLMs come out. Some of the challenges seem to be easily addressed by providing more in-domain data to the model which makes the trick easily broken.
2. For the question that are easy to LLM but hard to human, it is really hard to control whether the human will actually answer the questions without the help of computational tool or AI tool during the actual CAPTCHA check. Therefore, it is hard to really distinguish humans from LLMs by simply checking whether the memory type questions are answered correctly;

---

> ### Author Rebuttal · Authors · 2024-05-31
>
> Thank you for your thorough review of our paper. We appreciate your insightful feedback and would like to address the concerns raised regarding the reasons to reject our work.
>
> **Time-Wise Effectiveness of the Approach**: We acknowledge that the LLMs are evolving rapidly to gradually overcome these imposter detection questions. However, our method can also revolutionize at the same time. Following the proposed task taxonomy, the updated questions are still challenging for LLMs to solve them natively.  Questions like "Can we make 24 using only four 7's?" could easily fail the latest LLMs like GPT-4o.  Note that it represents a set of questions that need many search trials and possibly a mathematical proof showing that a solution does not exist.  Solving these questions will give new insights for the future development of LLMs. We will continuously update the question set, encouraging researchers from both sides to improve their methodology over time.
> The issue of distinguishing LLMs from humans exists and remains significant. Our work highlights this importance and proposes a different method to address it. Our approach is a step towards identifying and understanding the limitations of LLMs, and we envision that future research will build on this foundation to develop more sophisticated and adaptive detection methods.
>
> **Control Over Human Responses and Use of Computational Tools**: We recognize the concern regarding the possibility of human participants using computational or AI tools to answer questions that are difficult for them but easy for LLMs. To address this, we can rephrase questions to include contextual hints, which can deter the use of external tools. For instance, a question could be phrased as, "To your knowledge, list all the nations in the USA you know." Such phrasing encourages genuine human responses and makes it harder for individuals to rely on external aids.
>
> Additionally, implementing time constraints and monitoring response patterns can help mitigate the issue. Humans typically take longer time to derive answers to complex questions compared to LLMs, which can provide near-instantaneous responses. By analyzing not only the correctness of answers but also the time taken and the response style, we can enhance the robustness of our detection mechanism. Regular updates and refinements to the question sets will also help maintain their effectiveness over time.

---

> > ### Comment · Reviewer_fBF2 · 2024-06-04
> > **Response to Author's Rebuttal**
> >
> > Thanks for authors' responses. To me, my concern on the time-wise effectiveness of these set of questions still remain. Especially, the design of the questions heavily rely on manual effort, which makes it hard to scale up, whiich makes it hard to be deployed as a reliable Recaptha application. However, I do appreciate the value of the benchmark to study the distinguished behavior of humans and LLMs to the same set of questions and how to address the LLM's limited capability on them. I will slightly increase my score, but I personally would still not favor this method as a RECAPTCHA implementation.

---

> > ### Author Response · Authors · 2024-06-05
> >
> > Thanks for your consideration.  There will be competitive and possibly better approaches for RECAPTCHA in this new context.  But this is (likely) the first paper on this topic and it develops a new set of tests that target the key weakness and strength of LLMs.   For many of these tests, questions/answers can indeed be generated automatically on a large scale.   After we discussed this task with undergraduate students in an AI course,  in one day, they came up with the following new questions and many others.  It is not that hard when we have broader audiences.
> >
> > (1) If I have task <X> that takes 3 hours, task <Y> that takes 4 hours, and task<Z> that takes 2 hours; but I can do task <Z> at the same time as task <Y>. What is the minimum time I can allot to complete all these tasks?
> >
> > (2) Here's a simple grid labyrinth where you can move up, down, left, and right for 3 steps. What is the maximum sum of values you can collect? The letter 'P' marks your current location.
> > ```
> > 23 45 67 89 21
> > 43 56 78 90 12
> > 34 57  P 76 98
> > 10 09 87 65 43
> > 22 34 56 78 99
> > ```
> > Based on these templates, we can generate many different questions.  We sincerely hope through publication,  this paper can reach more audiences, and inspire new ideas along this line.  It will also help us test the boundary of LLMs.

---

### Decision · Program_Chairs · 2024-07-10

**Decision:**

Accept

**Comment:**

The paper proposes a novel method FLAIR to distinguish between human and LLM responses by asking questions that are either easy for LLMs but hard for humans, or vice versa. The researchers identify patterns in these questions that lead to significant performance differences between humans and LLMs. The proposed questions are validated against human users and a comprehensive set of LLMs, demonstrating the method's effectiveness in distinguishing between the two based on response accuracy.

Summary Of Reasons To Publish:
1. Innovative Approach: The paper introduces a simple yet effective method to capture the differing behaviors of humans and LLMs through well-designed questions.
2. Comprehensive Evaluation: The study provides detailed experimental results and extensive evaluations across various models and settings, adding to the robustness and credibility of the findings.
3. Clear Documentation: The question design and dataset generation are clearly explained, facilitating replication and further research.
4. Valuable Benchmark: The proposed method offers a valuable benchmark for evaluating LLMs' foundational skills that are inherently easy for humans, contributing to the field of AI safety and security.

Summary Of Suggested Revisions:
1. Time-wise Effectiveness: The authors should address the concern regarding the time-wise effectiveness of the proposed approach, considering the rapid advancements in LLM capabilities.
2. Manual Intervention Concerns: The paper should discuss the potential challenges of ensuring human participants do not use computational tools during CAPTCHA checks, particularly for questions easy for LLMs but hard for humans.
3. Framework for New Question Generation: A broader discussion on automated frameworks for generating new questions as LLMs overcome current limitations would enhance the paper. This includes addressing the scalability and adaptability of the question generation process.
4. Accuracy Calculation Details: Providing more detailed information on how accuracy was calculated, including the rationale for using median accuracy versus mean accuracy and the variance across trials, would improve clarity.

Overall Assessment:

Despite some concerns about the method's long-term effectiveness and practical application, the novelty and thoroughness of the approach justify its publication. Addressing the suggested revisions would further enhance the paper's impact and clarity.